# The Use of Non-Invasive Brain Stimulation for the Management of Chronic Musculoskeletal Pain: Fad or Future?

**DOI:** 10.3390/brainsci15070760

**Published:** 2025-07-17

**Authors:** Philippe Patricio, Hugo Massé-Alarie

**Affiliations:** 1Department of Psychology, McGill University, Montreal, QC H3A 1G1, Canada; philippe.patricio@mail.mcgill.ca; 2Center for Interdisciplinary Research in Rehabilitation and Social Integration, Quebec, QC G1M 2S8, Canada; 3Department of Rehabilitation, Faculty of Medicine, Université Laval, Quebec, QC G1V OA6, Canada

**Keywords:** chronic musculoskeletal pain, brain stimulation, review

## Abstract

This article aims to offer a broad perspective on the use of non-invasive brain stimulation (NIBS) techniques in the context of chronic musculoskeletal pain (CMP) conditions. While NIBS has demonstrated promising efficacy in certain chronic pain populations, its application in the management of CMP remains limited. This paper examines the current evidence supporting the use of NIBS for pain relief in CMP, the rationale and proposed mechanisms of action, the importance of patient selection, common methodological limitations in the existing literature, and the potential adverse effects of these techniques. The authors argue that the current evidence is insufficient to support widespread clinical adoption of NIBS for CMP. Advancing the field will require more rigorous study designs, with adequately powered and properly blinded randomized controlled trials. Additionally, future research should address the identification of potential responders to brain stimulation, conduct economic evaluations, and carefully assess the benefit–risk ratio before NIBS can be integrated into routine clinical practice.

## 1. Growing Popularity of Non-Invasive Brain Stimulation in Pain Management

Interest in non-invasive brain stimulation (NIBS) techniques has grown substantially over the past two decades, both in research and clinical settings. This rise in popularity is largely due to the versatility of these techniques. Since NIBS allows for the relatively targeted stimulation of nearly any cortical region, it can theoretically be applied to any health condition involving cortical dysfunction. For instance, NIBS has been widely used to treat deficits following stroke [1], as well as various neurodegenerative diseases (e.g., Parkinson’s disease, Alzheimer’s disease) [2,3], psychiatric disorders (e.g., anxiety disorders, obsessive–compulsive disorder, depression) [4], etc.

Clinically, repetitive transcranial magnetic stimulation (rTMS)—one of the most used NIBS techniques—is mainly utilized for the management of treatment-resistant major depressive disorder, which remains one of the few indications approved by regulatory agencies such as the U.S. Food and Drug Administration (FDA) [5]. In research, there is now scientific evidence supporting its effectiveness in treating neuropathic pain, although the effect is small [6,7]. NIBS techniques like rTMS and tDCS (transcranial direct-current stimulation) are also being investigated for other chronic pain conditions such as fibromyalgia and complex regional pain syndromes, with encouraging results [8,9,10], highlighting that research on NIBS for pain management has expanded in recent years. A simple PubMed search combining common NIBS terms with pain ([rTMS or tDCS or non-invasive brain stimulation] AND pain) revealed an average increase of thirteen additional publications per year over the past two decades when plotting a linear regression function from 2004 to 2024 (Figure 1).

This growing interest is not restricted to the field of research, even though regulatory bodies have approved NIBS for only a limited number of conditions. In Quebec, the use of certain NIBS techniques—particularly rTMS—for major depressive disorder is now covered by public health insurance. This has led to a rise in the number of private clinics offering NIBS-based treatments. The NIBS’ market is projected to grow from USD 1205.5 million in 2024 to an estimated USD 1980.15 million by 2032, worldwide [11]. Several public healthcare institutions also provide access to NIBS, including for pain management. The democratization of the use of NIBS in the clinical field is probably secondary to its recommendations as a potential treatment in clinical guidelines for the management of certain conditions such as depression [12] and neuropathic pain [13]. Nonetheless, evidence of efficacy for neuropathic pain does not mean that all pain conditions can respond well to NIBS. Indeed, despite the growing interest in the use of NIBS in the management of chronic pain, few studies have tested its effectiveness specifically for the management of chronic musculoskeletal pain (CMP), despite a call for studying its effectiveness for these conditions [14]. The main objectives of this perspective paper are to (i) present an overview of the current evidence on the effectiveness of NIBS for the management of CMP, (ii) discuss the rationale for the use of NIBS in pain management and if subgroups of patients with CMP may respond best, and (iii) present future steps that are crucial to eventually encourage (or discourage) the wider usage of NIBS in the clinical field for the management of CMP. Before addressing these elements, it is important first to clearly define CMP and NIBS techniques.

## 2. The Definition of Chronic Musculoskeletal Pain (CMP)

In 2021 in the US, CMP was the most prevalent chronic pain condition [15]. It was initially defined as pain affecting the muscles, bones, tendons, or joints for more than 3 months [16]. Considering that this definition did not consider the underlying mechanisms of pain according to the latest scientific evidence, the 11th edition of the International Classification of Diseases (ICD-11) has now separated CMP into primary and secondary CMP [17]. The former can be defined as long-lasting pain in muscles, bones, or joints that causes emotional distress or disability that is not explained by a clear disease or injury, whereas the latter encompasses pain in the same areas but caused by identifiable conditions like inflammation, structural damage, or neurological disorders [17]. As this new classification is still seeing debate as to which conditions should enter each category [18,19,20], and for clarity in the present paper, we decided to include the following conditions under the concept of CMP: chronic primary spinal pain (i.e., neck, thoracic, and lower back), myofascial pain syndrome (a pain syndrome that affects muscles and fascia, sometimes involving myofascial trigger points) [21], and CMP associated with structural changes [17] (i.e., secondary pain, e.g., osteoarthritis [OA], rotator-cuff-related shoulder pain, sub-acromial pain syndrome, and patellofemoral pain syndrome). Fibromyalgia and complex regional pain syndrome (CRPS) type 1 or 2 are not considered as CMP as these conditions belong to other chronic primary pain categories according to the ICD-11 (chronic widespread pain and CRPS) [22].

## 3. Types of Non-Invasive Brain Stimulation Techniques

NIBS encompasses a wide array of technologies that stimulate the brain and modulate its activity without the need for surgery or invasive procedures, thereby distinguishing them from invasive methods such as deep brain stimulation [7]. NIBS techniques are commonly divided into two categories: magnetic stimulation and electrical stimulation [23].

### 3.1. Magnetic Stimulation

Based on the work of Barker and the introduction of single-pulse magnetic stimulation of the brain in 1985 [24], rTMS has emerged as one of the major NIBS techniques for the treatment of chronic pain since its first use in the early 1990s [25]. It is based on the induction of electric currents in neurons by using a magnetic field. It can modify brain function in local and remote areas, with low-frequency stimulations (≤1 Hz) lowering cortical excitability and high-frequency stimulations (≥5 Hz) increasing it. The full mechanisms underlying the analgesic effect of rTMS remain to be fully understood (for more details, see Section 5). High-frequency rTMS over the primary motor cortex (M1) has been shown to be effective in treating various chronic pain conditions, with most evidence supporting its use in neuropathic pain [6].

### 3.2. Electrical Stimulation

Electrical stimulation of the brain covers several techniques, including tDCS and cranial electrotherapy stimulation (CES), both implying the use of a low-intensity current (≤2 mA). The principle of action of tDCS is based on modifying the resting potential of the stimulated neurons, either by hyperpolarization or depolarization, depending on the direction of the current relative to the orientation of the axons [10]. Anodal tDCS (a-tDCS) increases excitability in the targeted brain region, whereas cathodal stimulation has the opposite effect, reducing cortical excitability [26]. This has been confirmed by studies combining functional near-infrared spectroscopy and a-tDCS, showing an increase in cortical activation at the site of stimulation [27]. Pain reduction is often achieved by using a-tDCS over M1 [10]. For CES, electrodes are placed on the mastoid processes or clipped to the earlobes. This technique is thought to have an analgesic effect through the modulation of brain regions involved in pain processing, an increase in pain-inhibiting neurotransmitters like serotonin and dopamine, and the activation of endogenous opioid pathways [28]. A less-studied electrical stimulation technique called transcranial alternating-current stimulation (tACS) delivers an oscillatory electrical current that alternates between positive and negative voltages to modulate the brain’s endogenous neural oscillations [29]. In the context of pain, tACS has been investigated based on evidence suggesting that pain can disrupt alpha-band oscillations [30,31] and that such disruptions are correlated with pain intensity in certain clinical populations [32].

## 4. Evidence of Efficacy in CMP

A PubMed and Google Scholar search was performed in May and June 2025, combining terms for NIBS (e.g., rTMS, transcranial magnetic stimulation, tDCS, CES, and non-invasive brain stimulation) and musculoskeletal conditions (e.g., myofascial pain syndrome, low back pain, chronic pain, and osteoarthritis). To provide a broad assessment of the available evidence, we prioritized systematic reviews and meta-analyses when available. If not mentioned in the systematic reviews retrieved, individual studies were also integrated into the present paper. References from included articles were also reviewed. Table 1 reports on the different studies (systematic reviews and original randomized controlled trials) included in this perspective paper.

### 4.1. Chronic Primary Spinal Pain

Studies investigating the effectiveness of NIBS for chronic primary spinal pain have predominantly focused on non-specific chronic low back pain (CLBP). A recent systematic review assessing the efficacy of NIBS in CLBP included twelve randomized controlled trials (RCT)—eight on tDCS, two on rTMS, one on CES, and one on tACS [33]. The only statistically significant meta-analysis results reported in the review was a reduction in pain following a single NIBS session compared to a sham (i.e., a simulated treatment to blind patients and/or therapists, to account for contextual effects), based on very-low-quality evidence (standard mean difference (SMD): −0.47; 95% CI: −0.65, −0.29; *p* < 0.001). Regarding repeated sessions of stimulations, no significant differences were observed for short- and mid-term pain or when NIBS was combined with co-interventions (i.e., TENS, cognitive behavioral therapy, and physical therapy), with moderate-quality evidence.

According to this review, among the individual NIBS techniques, tDCS was the most extensively studied. Its lack of efficacy appears to be confirmed by a systematic review and meta-analysis focusing exclusively on this method, which mostly targeted M1 [34]. Moreover, a recent small study (n_total_ = 20) not included in the previous meta-analyses also tested the effect of combining pain education and tDCS over the dorsolateral prefrontal cortex (DLPFC) compared to education and a sham. It did not report an effect on pain, but on pain catastrophizing, a factor of poor prognosis [35].

The evidence appears less definitive when examining individual studies on rTMS, CES, and tACS. The two studies on M1-rTMS included in the previously mentioned review [33] yielded significant results regarding pain but were deemed to be at high risk of bias [36,37]. The first one used an open-labeled design where participants (n_total_ = 67) knew their allocated group (placebo or active rTMS) [36], whereas the second one evaluated pain after just one session of stimulation in only 10 participants [37]. However, the results of our recent adequately powered factorial trial on rTMS associated or not with motor control exercise (n_total_ = 140) found no effect of this combination over sham stimulation alone [38]. For electrical stimulation, eight sessions of CES were found to be ineffective in reducing pain (n_total_ = 33) [39], whereas tACS demonstrated positive effects following a single session of stimulation in a pilot study (n_total_ = 20) [40]. Overall, the evidence on NIBS modalities other than tDCS for CLBP underscores the need for more rigorous and well-designed RCTs. Aside from CLBP, one study evaluated the effect of 16 a-tDCS sessions over M1 with stabilization exercises in chronic neck pain [41]. A significant difference in pain was observed between the active (n = 10) and sham (n = 9) stimulation groups, both of which received adjunct exercise therapy.

### 4.2. Myofascial Pain Syndrome

Both tDCS and rTMS have been used in association with other treatments or as a stand-alone therapy to try to alleviate pain in this condition. For tDCS, when combining standard care (stretching, ultrasound, and hot packs) with active (n = 16) or sham (n = 15) a-tDCS for 5 days over M1, Sakrajai et al. [42] found a significant effect favoring the active group immediately post-treatment and at a 1-week follow-up, but not at 3- and 4-week follow-ups. Another pilot study [43] compared the efficacy of combining trigger point injections of lidocaine either with tDCS over M1 (n = 8), DLPFC (n = 7), or sham tDCS over M1 (n = 6) for 5 consecutive days. No differences were found between the three groups at the end of the treatment period.

For rTMS, a small RCT included 24 participants randomized to 10 sessions of a placebo or real rTMS over M1 [44]. They found significant pain reduction in favor of the active group, with a 30.21% (95% CI, −39.23 to −21.20) reduction in pain compared to the placebo arm 12 weeks after the treatment period. Another study by Medeiros et al. [45] used a four-group factorial design (n_total_ = 46) with M1-rTMS (real or sham) and deep intramuscular stimulation (DIMST—real or sham). The authors reported significant main effects of treatment and time on pain intensity, with all active treatment groups exhibiting significantly lower pain levels compared to the sham-rTMS + sham-DIMST group. However, they did not report the interaction between time and treatment, limiting conclusions about whether any one intervention was more effective than the others.

### 4.3. Osteoarthritis

Most studies evaluating the effects of NIBS on pain reduction in OA have used a-tDCS over M1. A recent systematic review published in 2023 [46] included 14 studies encompassing a total of 740 individuals with knee OA, examining the efficacy of a-tDCS either as a standalone intervention or in combination with other non-pharmacological treatments. Six studies were included in a meta-analysis comparing tDCS alone with sham stimulation, showing a significant effect of tDCS over the sham less than one week post-intervention (SMD: −0.47; 95% CI −0.89, −0.05; *p* = 0.03). However, the medium- (≥1 to 6 weeks post-intervention) and long-term effects (≥6 weeks post-intervention) did not show a significant effect on pain. In contrast, the meta-analysis of five studies on the effect of tDCS used as an adjunct to other therapies (i.e., quadriceps strengthening, TENS, intramuscular electrical stimulation, and conventional physical therapy) produced more consistent results. These effects were significant at a short-term follow-up (SMD: −0.86; 95% CI: −1.20, −0.52; *p* < 0.001), as well as at medium- and long-term follow-up periods. However, it is essential to note that the overall quality of evidence for these meta-analyses was low, due to the high risk of bias and the small sample sizes of the individual studies included. CES was used in a pilot study to treat knee and hip OA, but the results did not show any significant differences between the active and sham groups [47].

### 4.4. Other Joint Pain

A pilot study by Larrivée et al. [48] found that adding a single 20 min session of a-tDCS two weeks after a subacromial corticosteroid injection was not better than sham tDCS and an injection in reducing pain in patients with subacromial pain syndrome (n = 12 per group). Belley Fournier et al. [49] compared five sessions of sham or active tDCS in association with eight sessions of sensorimotor training (n = 20 per group) over 6 weeks for rotator cuff tendinopathy and found no significant improvement in disability between groups. In an RCT involving 28 women with patellofemoral pain, Rodrigues et al. [50] stated that 12 sessions of a-tDCS combined with resistance training significantly reduced pain perception. However, no significant between-group differences were observed when the results were compared to the group receiving sham tDCS and resistance training, leading to a potential misinterpretation of within-group significance as evidence of a treatment effect.

**Table 1 brainsci-15-00760-t001:** Studies included in the perspective article.

Specific Condition Authors, Year	NIBS Techniques (Target)	Study Design and Sample Size	Results	Power Calculation	Assessment of Blinding	Prospective Registration
Chronic Primary Spinal Pain						
CLBP Patricio, 2021 [33]	All NIBS techniques (tDCS, rTMS, tACS, and CES)	Systematic review with meta-analysis Twelve studies included (n total = 492)	- A single session of NIBS can reduce pain compared to placebo NIBS (very-low-quality evidence) - Repeated sessions of NIBS have no effect in the short and medium term on pain compared to placebo NIBS (moderate-quality evidence) - Not more effective when combined to a co-intervention (moderate-quality evidence)	N/A	Yes for 4 out of the 12 studies included	N/A
CLBP Alwardat, 2020 [34]	tDCS	Systematic review with meta-analysis Nine studies included (n total = 411)	- The pooled analysis showed no statistically significant improvements in pain in favor of tDCS over M1 compared to sham tDCS (no assessment of the overall quality of the evidence)	N/A	N/A	N/A
CLBP Patricio, 2023 [38]	rTMS (M1)	Factorial RCT (n total = 140)	- The 3 groups, rTMS, rTMS + exercise, and sham rTMS + exercise, did not outperform the sham rTMS group for pain reduction	Yes	Yes	Yes
CLBP Alcon, 2025 [35]	tDCS (DLPFC)	RCT (n total = 20)	- The combination of tDCS and education was not more effective than placebo tDCS and education to reduce pain	No	No	Yes
Neck pain Dere, 2025, [41]	tDCS (M1)	RCT (n total for the groups of interest = 19)	- The combination of tDCS + exercise was more effective in reducing pain than placebo tDCS + exercise	Yes	No	No
Myofascial Pain Syndrome (MPS)						
MPS Sakrajai, 2014 [42]	tDCS (M1)	RCT (n total = 31)	- tDCS with exercise was more effective than sham tDCS with exercise to reduce pain in the short term (1-week follow-up), but not at 3- and 4-week follow-ups	No	No	No
MPS Choi, 2014 [43]	tDCS (M1 and DLPFC)	RCT (n total = 24)	- tDCS over the DLPFC, over M1, or sham tDCS did not show any between-group differences in pain reduction	No	No	No
MPS Dall’Agnol, 2014 [44]	rTMS (M1)	RCT (n total = 24)	- rTMS significantly reduced pain compared to sham rTMS up until 12 weeks after the conclusion of the interventions.	Yes	No	Yes
MPS Medeiros, 2016 [45]	rTMS (M1)	Factorial RCT (n total = 46)	- The 3 groups, rTMS + DIMST, sham-rTMS + DIMST, and rTMS + sham-DIMST, showed lower pain compared to sham-rTMS + sham-DIMST. However, the group x time interaction was not reported	Yes	No	Yes
Osteoarthritis						
Knee osteoarthritis Dissanayaka, 2023 [46]	tDCS	Systematic review with meta-analysis 14 studies included (n total = 740)	- tDCS was significantly better at reducing pain than sham tDCS in the short term (less than a week post-intervention) but not at longer follow-ups (low-quality evidence) - tDCS combined with other therapies was better than sham tDCS combined with other therapies at all follow-up durations (low-quality evidence)	N/A	N/A	N/A
Hip/knee osteoarthritis Katsnelson, 2004 [47]	CES	RCT (n total = 64)	- tDCS using symmetric or asymmetric waveforms was not better than sham tDCS to reduce pain	No	No	No
Other Joint Pain						
Shoulder pain Larrivée, 2021 [48]	tDCS (M1)	RCT (n = 38)	- One session of tDCS following a corticosteroid injection did not reduce pain more than sham tDCS and an injection or an injection alone	No	No	No
Shoulder pain Belley, 2018 [49]	tDCS (M1)	RCT (n = 40)	- No significant effect of combining tDCS and sensorimotor training compared to sham tDCS + sensorimotor training on symptoms and function	Yes	Yes	Yes
Patellofemoral pain Moraes Rodrigues, 2021 [50]	tDCS (M1)	RCT (n = 28)	- No between-group difference on pain level was found between tDCS + resistance training and sham tDCS + resistance training groups	No	No	No

CES: cranial electrotherapy stimulation; CLBP: chronic low back pain; DIMST: deep intramuscular stimulation therapy; DLPFC: dorsolateral prefrontal cortex; M1: primary motor cortex; MPS: myofascial pain syndrome; N/A: not assessed in the systematic review; NIBS: non-invasive brain stimulation; RCT: randomized controlled trial; rTMS: repetitive transcranial magnetic stimulation; tACS: transcranial alternating current stimulation; and tDCS: transcranial direct-current stimulation.

### 4.5. General Overview of the Evidence

So far, the effects of NIBS for the treatment of CMP appear small at best, with limited and mixed results. For CLBP, NIBS showed no consistent benefits from repeated sessions or combinations with other therapies. CES and tACS presented some promising results but suffer from small, biased studies, while the recent results on rTMS seem to show that it lacks efficacy. For myofascial pain syndrome, inconsistent results were found and often limited by low-quality, small sample-sized trials. On the other hand, the evidence for osteoarthritis, especially tDCS, appears promising but requires confirmation with high-quality RCTs. Finally, results for other joint pain were mostly inconclusive. The proportion of original RCTs reporting power calculation, blinding efficacy, and prospective registration are presented in Figure 2. To better understand this modest effect, we reviewed the proposed mechanisms underlying the reduction in pain perception in both pain-free and chronic pain populations. Note that few studies specifically tested physiological mechanisms in CMP conditions.

## 5. Rationale of the Use of NIBS in Chronic Musculoskeletal Pain

The rationale for using NIBS depends on the stimulation site and the specific parameters employed. For pain treatment, many studies have targeted M1 to induce analgesic effects. While targeting M1 may seem counterintuitive, there is strong evidence supporting its efficacy compared to a placebo or other stimulation sites [51]. M1 stimulation was implemented following the success of deep brain stimulation for pain relief, for which electrodes are usually implanted in this cortical area [52]. It has been suggested that analgesia induced by deep brain stimulation and rTMS over M1 may share common mechanisms (see [52] for a review). Although M1 is not typically considered a key structure in descending pain modulation, the effectiveness of its stimulation to reduce pain (experimental and clinical) raises important questions about the underlying mechanisms of its analgesic effects. It has been proposed that the connectivity between M1 and the thalamus (cortico-thalamic connections), and between the thalamus and the periaqueductal gray matter (PAG—thalamo-mesencephalic connections), may serve as the anatomical substrate through which NIBS exerts its analgesic effects. The PAG is situated in the brainstem and—with the rostral ventral medulla—plays an important role in descending pain modulation [53]. Moreover, the structural and functional connectivity between these two areas seems important for pain relief to occur following brain stimulation. Goto et al. [54] observed that individuals suffering from central post-stroke pain and with reduced integrity of the thalamocortical tract (<50% compared to the uninjured side) did not benefit from rTMS. In line with this, individuals with fibromyalgia that had the strongest functional connectivity between M1 and the thalamus before tDCS presented the largest improvement at the end of the intervention [55]. However, these findings are based on small sample sizes and require replication to confirm their physiological validity. Another hypothesis suggests that M1 stimulation-induced hypoalgesia may be due to the release of endogenous opioids. For example, the administration of naloxone in pain-free humans—an µ-opioid receptor antagonist—has been shown to block the analgesic effects of M1-rTMS [56].

NIBS is also frequently applied to the DLPFC. Many studies provide the same rationale for stimulating the DLPFC versus M1; the DLPFC is rich in opioid receptors [57] and its stimulation reduces pain sensitivity in pain-free humans [58]. However, the abolition of rTMS-induced hypoalgesia in response to naloxone was not observed in all studies, suggesting the contribution of both opioid and non-opioid receptors [56,59]. Another rationale is based on the recognized efficacy of left DLPFC stimulation to treat psychological disorders such as major depression and anxiety disorders [60]. The presence of psychiatric disorders (and other negative cognitive and emotional factors) is associated with long-term disability in many chronic pain conditions [61,62]. Thus, it has been suggested that rTMS could modulate the excitability of the DLPFC—also involved in cognition and emotion processing—and may limit the impact of these psychological factors that negatively influence recovery in chronic pain patients [63,64]. These factors include fear of movement and pain, catastrophizing, anxiety, and depressive symptoms [65,66]. Therefore, targeting the DLPFC may influence both pain perception and the cognitive–emotional factors that may contribute to hindering recovery. Other cortical regions, such as the somatosensory cortex [67], have also been explored, but studies remain limited and the effectiveness of targeting these areas is not well established.

Some authors have also suggested that NIBS could enhance the brain’s sensitivity to conventional treatments through the improvement of neural plasticity and learning [68]. Indeed, motor learning processes are accompanied by variations in cortical excitability and changes in synaptic efficacy. Considering that the effects of non-invasive brain stimulation (NIBS) depend on N-methyl-D-aspartate (NMDA) receptors, stimulation-induced changes in cortical excitability may interact with motor learning processes, potentially enhancing them through the strengthening of NMDA receptor activity [69]. Most of the research in this area has focused on post-stroke motor recovery [70], and evidence for CMP is still lacking. Synergistic mechanisms directly related to pain have also been proposed between exercise and brain stimulation [71]. This combination may work synergistically to modulate pain through both ascending and descending pathways. Exercise likely generates ascending sensory input, activating the endogenous opioid system [72]. Concurrently, rTMS or tDCS may enhance sensorimotor–thalamic connectivity and activate the periaqueductal gray matter, thereby engaging the endogenous descending pain modulation system. A meta-analysis found a positive effect of combining exercise with rTMS or tDCS to reduce chronic pain compared to exercise and sham NIBS [71], but the overall quality of the evidence was not assessed, and the chronic conditions studied were not restricted to CMP (e.g., fibromyalgia, CRPS type I).

In summary, the neural mechanisms proposed for the use of NIBS in the management of chronic pain are based on its capacity to reduce pain intensity by the activation of the pain modulation system (both M1 and the DLPFC), to impact psychological factors associated with poor prognoses by the modulation of the DLPFC, and/or to prime the brain to better respond to conventional treatments. The first two proposed mechanisms may be interpreted in two opposite ways. First, NIBS can activate the pain modulation system—eliciting diffuse widespread analgesia—hence improving the symptoms in patients with chronic pain. This first rationale implies that the pain modulation system is responsive in chronic pain populations and produces transient effects that need to be repeated to be maintained in the long run. If so, the benefits secondary to NIBS need to overcome the disadvantages inherent in its use (potential risks, cost of treatment, etc.; see Section 8 on risks and benefits). If not, a simple analgesic method such as transcutaneous electrical nerve stimulation (TENS)—which is also known to activate the pain modulation system—could be as beneficial as NIBS with lower costs and risks. For example, Hazime et al. [66] reported no difference between TENS and tDCS for the treatment of CLBP. Future studies comparing NIBS with less risky and costly alternatives, such as TENS for pain are a necessity. A second rationale is that NIBS can “re-establish” the proper functioning of an altered cerebral pain system in patients living with chronic pain [73]. This second rationale seems to contradict the first one, for which the pain modulation system needs to function properly. It also implies that (i) NIBS can elicit long-lasting plasticity (i.e., for weeks and months) within the brain, (ii) these changes mediate the improvement in pain and/or physical function (i.e., are associated with improvement), and, importantly, (iii) the brain is truly “altered” in chronic pain. The fact that repeated NIBS sessions are necessary to maintain the effects seems to support the first rationale, whereas the abundance of evidence of altered brain processing and connectivity, and reduced efficacy of descending pain modulation in chronic pain, seems to contradict it. Therefore, the mechanisms underlying the effects of NIBS in chronic pain remain to be fully deciphered and could lie between the latter two alternatives. This leads to the next crucial question: “Do patients with CMP have an alteration of their pain modulation system?”

## 6. Different Pain Mechanisms, Different Responses to Treatment?

As mentioned previously, the ICD-11 classified patients with CMP as having primary or secondary pain. Considering that the primary pain category cannot be explained by a lesion or a disease, its definition seems to overlap with nociplastic pain, which is defined as *“pain that arises from altered nociception not fully explained by nociceptive or neuropathic pain mechanisms”* [74]. In contrast, secondary pain is considered a symptom of an underlying lesion or disease, corresponding more to nociceptive pain, which is defined as *“Pain that arises from actual or threatened damage to non-neural tissue and is due to the activation of nociceptors”* [75]. Some conditions considered as being predominantly nociplastic, such as fibromyalgia and chronic widespread pain [76], are outside the scope of this perspective article, as is neuropathic pain. Nonetheless, it has been proposed that some CMP conditions, like non-specific CLBP, may evolve on a spectrum from predominantly nociceptive to predominantly nociplastic pain [77,78]. These pain mechanisms are not considered to be mutually exclusive, meaning that the chronic pain from a specific patient can have contributions from all types of pain (e.g., nociception contributes to CMP, but with persistent peripheral and/or central amplification). Note that there are some inconsistencies between the ICD-11 and pain mechanism classifications despite these convergences. For example, while the ICD-11 classifies non-specific CLBP as a primary pain condition, multiple pain mechanisms may contribute to an individual patient’s experience (nociceptive, neuropathic, nociplastic, or mixed) [78]. In this perspective paper, we argue that all musculoskeletal pain is not the same, and different mechanisms may contribute to a different extent to each patient’s profile. The differences between pain mechanisms are crucial for the treatment of CMP using NIBS. Indeed, patients with predominantly nociplastic CMP are thought to exhibit altered processing of pain in the central nervous system (e.g., increased functional connectivity between the nucleus accumbens and the prefrontal cortex in CLBP) [79,80], whereas those with predominantly nociceptive CMP should present with minimal or no alteration within the central nervous system. Therefore, if we return to the underlying mechanisms of the effects of NIBS in consideration of the predominantly CMP mechanisms, the response to NIBS could be hypothesized in two opposite directions:Patients with predominantly nociplastic CMP should respond less to NIBS because of improper functioning of their pain modulation system, while patients with predominantly nociceptive CMP could benefit from NIBS via the transient activation of the pain modulation system (similarly to using medications, TENS, or any other analgesics). For the latter, repeated sessions should be needed to maintain pain relief; any cessation of NIBS sessions would result in the loss of gains.Patients with nociplastic CMP should respond better to NIBS because it would “re-establish/re-set” the functioning of their pain system, while patients with nociceptive CMP may benefit from a transient effect. Repeated sessions may be necessary to elicit long-lasting neuroplasticity, but may be stopped once neuroplasticity is installed without losing the gains.

In the literature, pain mechanisms are rarely considered; usually, patients with CMP conditions (e.g., chronic neck pain) are considered a homogeneous group recruited based on the simple criteria of duration (>3 months) and/or frequency (>50% of the days in the last 3 months) [81]. Nonetheless, it remains difficult to diagnose if a given patient has predominantly nociceptive or nociplastic CMP, given that these mechanisms are not mutually exclusive. Efforts are currently being made to build diagnostic tools to better discriminate between these pain types [78,82], and their validation is crucial for advancing the field.

In the meantime, moderation analyses can be used to identify features associated with different pain mechanisms that may predict those who will favorably respond to NIBS, but not to sham NIBS. For example, in our latest RCT, we identified only one moderator of the intervention, which was the score for the central sensitization inventory (CSI) [83]. This questionnaire was initially developed to assess somatic symptoms, and emotional complaints often present in individuals with predominantly nociplastic pain [84]. Although this questionnaire was developed based on symptomatology such as fibromyalgia, chronic fatigue, and irritable bowel syndrome, it has been used—in combination with other features—to identify patients with nociplastic CLBP [85]. We established the rationale that participants with higher scores for the CSI will benefit more from rTMS according to the second hypothesis (those with potentially nociplastic CMP may respond better to NIBS). However, we observed the opposite results, suggesting a negative interaction between higher scores for the CSI and active rTMS [83]. Participants in this group improved less than all the other subgroups, including both sham rTMS groups (with high and low CSI scores). Although this result needs to be validated in external studies, it suggests that, for CMP conditions such as CLBP, rTMS could interact negatively with potentially altered brain functioning. The latter results contrast with some predominantly nociplastic pain conditions, such as fibromyalgia, that usually respond to NIBS [86]. Methodological factors may—at least in part—explain the latter discrepancy in NIBS and pain research, especially the control of the placebo and contextual effects that are deemed to be large.

## 7. Methodological Considerations: Blinding Efficacy and Placebo/Contextual Effects Secondary to NIBS

When evaluating the evidence for the efficacy of NIBS techniques in managing CMP, it is obvious that, although a few trials were adequately powered, many suffer from having a small sample size (Table 1). As smaller trials tend to report more significant results [87], this highlights the need for future studies to include appropriate a priori sample size calculations and pre-registration on clinical trial registries. While smaller trials can be pooled in meta-analyses, substantial variability in the number of sessions, co-interventions, stimulation targets, follow-up durations, and stimulation parameters contributes to heterogeneity and reduces the certainty of the evidence. Additionally, few of the studies referenced in this paper assessed or reported the effectiveness of their blinding procedures. This is particularly important in studies targeting M1, where stimulation can sometimes induce visible motor responses, making it difficult to maintain blinding for both participants and the technicians administering the treatment. Participants who suspect that they are in the active group may experience an enhanced placebo effect, while those who believe they are in the sham group will have lowered expectations and effects on pain. This implies that studies should also report whether participants are naïve to rTMS, as it can influence their ability to guess group allocation. It is well known that pain is one of the outcomes most influenced by the placebo effect [88], particularly higher for procedures/devices than medications [89]. Hence, the use of NIBS in clinical (and research) settings can have a large effect size on pain-related outcomes, which need to be considered in the broader perspective of contextual effects. This is particularly true concerning rTMS, which combines numerous contextual factors known to potentially enhance a placebo effect, as outlined by Burke et al. [90]. They mentioned the use of an impressive rTMS device applied to the head, often accompanied by MRI-guided neuro-navigation with hands-on setup and calibration, and the visual feedback from electromyogram recordings that can influence expectations. Patients also engage in extended interactions with technicians and typically receive treatment in specialized clinics or academic medical centers. Therefore, when assessing NIBS efficacy for pain, it is of the utmost importance to use a sham group and to measure blinding efficacy in research settings to avoid overestimating the specific effects of NIBS on pain.

For rTMS, an ideal sham condition would avoid any meaningful brain stimulation while preserving the same visual, acoustic, and sensory experiences. To achieve this, the current gold standard is the use of a sham coil that prevents the magnetic field from reaching the cortex while maintaining all other characteristics of a real coil. Nonetheless, it is currently very challenging to perform a RCT that blinds participants and technicians in research because of the complexity of blinding the technician to the nature of stimulation (e.g., muscle twitch in the face produced by DLPFC-rTMS or the need/choice to confirm stimulation parameters’ “hotspot”/motor threshold for M1-rTMS). In contrast, it is easier to perform an RCT blinding both the participants and the technician using tDCS. Active or sham tDCS can more easily blind the provider by entering a code in the tDCS software for randomization. Sham tDCS often consists of a ramp-up/ramp-down protocol, in which real electrical stimulation is applied briefly (~30 s) at the beginning of the session and then discontinued without the participant’s knowledge [7]. This strategy is believed to work, and the tingling perceived during active tDCS usually fades out during the first seconds of stimulation [91]. However, if this strategy works well for a stimulation intensity of 1 mA [92], it seems less effective when used at intensities of 2 mA [93]. Consequently, future studies should employ state-of-the-art sham control, if possible, but also always assess and report the efficacy of blinding in RCTs.

## 8. Balancing the Benefits and the Risks of the Use of NIBS in CMP

Based on the evidence reviewed in this paper, the efficacy of NIBS for CMP remains highly uncertain except for tDCS in CLBP, where current evidence suggests a lack of efficacy [33,34]. The clinical application of NIBS should therefore be weighed carefully against its potential risks. Although NIBS is associated with considerably fewer adverse effects than invasive brain stimulation, all forms of NIBS are linked to minor or transient side effects, including headaches, scalp discomfort, and dizziness [94]. The most serious event reported using NIBS is the induction of a seizure by rTMS [95]. The risk is considered low and can be further reduced by controlling for the presence of risk factors associated with rTMS-provoked seizures, like a history of epilepsy, previous brain damage, or medication known to lower seizure threshold [95]. However, it is important to note that this major adverse effect can occur without any of those factors, even if the stimulation parameters are within the safety guidelines [96]. A Cochrane review on the effect of NIBS on chronic pain highlighted that out of the 94 trials included, 24 did not report any information regarding adverse events, and many of the remaining studies did not adequately report details [7]. Taken together, even if most risks are minor, high-quality evidence is missing for NIBS’ real-world implementation in the context of musculoskeletal pain conditions. Additionally, the lack of standardized protocols and the considerable variability in individual responses present significant barriers to clinical application. High-quality RCTs are needed to establish efficacy, optimize stimulation parameters, and identify patient subgroups most likely to benefit from NIBS, if effective at all. We argue that it is time for studies evaluating the interest of NIBS in treating CMP to move from small trials commonly published in every new and exciting field to adequately powered, blinded, and designed trials with the full reporting of outcomes [97]. This will probably diminish the effect size inflation caused by small studies [98], as shown in tDCS for CLBP. However, it is a necessary step to advance the field and determine whether NIBS deserves a place in the clinician’s toolbox for managing CMP or may ultimately prove unsuitable for clinical application. Additionally, future high-quality RCTs should include an economic evaluation of NIBS, which is particularly important for determining if its benefits overcome the higher costs of these technologies. Finally, if NIBS is found to be effective for some patients with CMP, it will be crucial to encourage its clinical applications in alignment with best practice guidelines for pain management, i.e., to complement high-value care, such as physical and psychological therapies that are recommended by most guidelines (e.g., Zhou et al., 2024) [99].

## 9. Conclusions

This perspective article underscores that, despite growing interest in NIBS techniques for treating pain conditions such as CMP, their efficacy remains largely uncertain. There is a pressing need for rigorously designed, adequately powered, and properly blinded RCTs. Furthermore, a deeper understanding of the underlying mechanisms of action, optimal stimulation protocols and parameters, and criteria for identifying patients with CMP most likely to benefit from NIBS is essential before these techniques can be widely implemented in clinical practice, with careful consideration of their benefit–risk ratio.

## Figures and Tables

**Figure 1 brainsci-15-00760-f001:**
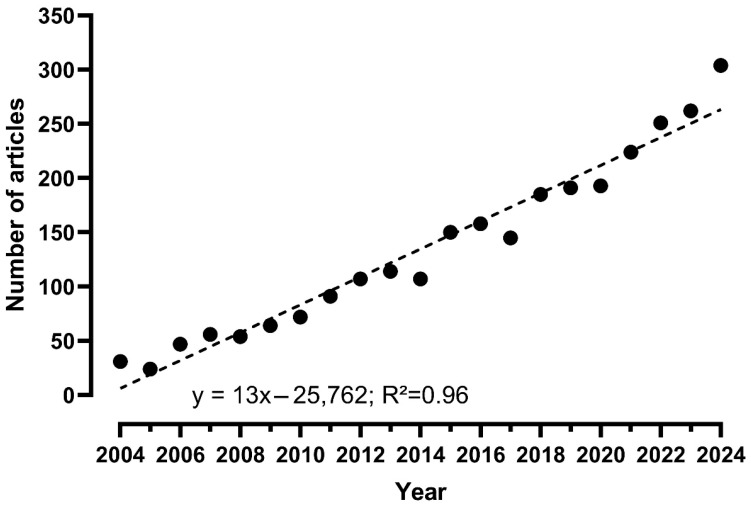
Evolution of the number of articles published in the field of non-invasive brain stimulation and pain in the last 20 years.

**Figure 2 brainsci-15-00760-f002:**

Number of original RCTs that reported a power calculation, a blinding assessment, and a prospective registration. Note that this was performed only for papers not included in meta-analyses, except for one meta-analysis that reported blinding assessment in the text. Therefore, the percentages are not exhaustive but are intended to show trends.

## Data Availability

No new data were created or analyzed in this study.

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
