# Peer review of "The Use of Non-Invasive Brain Stimulation for the Management of Chronic Musculoskeletal Pain: Fad or Future?"

_brainsci, 2025, doi:10.3390/brainsci15070760_

Round 1
Reviewer 1 Report
Comments and Suggestions for Authors
-In the first paragraph you mention that NIBS can theoretically be applied to any health condition involving cortical dysfunction and you use as examples various neurodegenerative diseases (e.g., Parkinson’s disease, Alzheimer’s disease) and psychiatric disorders (e.g., anxiety disorders, obsessive-compulsive disorder, depression) which do not have cortical dysfunction as their preliminary cause. You could mention the mechanism of action (proven or hypothetical) for each category of disease and provide data again for each one based on your references.
-In the second paragraph you have to explain first the acronym tDCS in full words before using it.
-In line 125-126, once you have previously described the term tDCS, you can use just the acronym.
-In line 132 you should explain what M1 means (primary motor cortex?).
-The sentence in line 142-143 “ The next section will review… of CMP” is redundant, could be omitted.
-In line 148 the term MSK means musculoskeletal? It would be better for readers to write the full word.
-In section 4 the use of the term “sham” is quite controversial because it is not explained early in the paper but only in the final chapters readers can conclude what it means. Please explain better from the beginning what it refers to.
-In section 4, Evidence of efficacy in CMP, the absence of tables is not helpful to summarize the findings from the literature review. Just mentioning few facts about each paper without a structured manner is not helping audience to draw specific conclusions.
Reviewer 2 Report
Comments and Suggestions for Authors
Thank you for the opportunity of reviewing the manuscript by Patricio and Masse-Alarie, titled "The use of non-invasive brain stimulation for the management of chronic musculoskeletal pain: fad or future?"
The manuscript is well written and easy for the readers to follow. The content provides a great overview on the current application of NIBS for chronic muscusloskeletal pain, I only have some minor comments for the authors to consider:
- Please provide more details on the database search and the date of the last search performed. While this is not a systematic review, the authors aim to provide an overview of current evidence for NIBS. It's important that readers know how current the evidence is. Such information can be provided as a supplementary document.
- While the neural mechanisms underlying NIBS were discussed, findings of the brain hemodynamics (recorded by fNIRS) related to tDCS were not mentioned. Including findings from broader neuroimaging results such as fNIRS studies would be complementary to the overview of the brain mechanisms related to NIBS. For example: Ronak Patel, Aleksander Dawidziuk, Ara W. Darzi, Harsimrat Singh, Daniel R. Leff, "Systematic review of combined functional near-infrared spectroscopy and transcranial direct-current stimulation studies," Neurophoton. 7(2) 020901 (25 June 2020) https://doi.org/10.1117/1.NPh.7.2.020901
- When applying NIBS for chronic pain management, in addition to being a stand alone approach, another approach is "priming" - using NIBS to augment the effects of other conventional interventions (i.e., exercise) to provide synergistic effects. Could the authors provide more dicussion about NIBS as stand alone vs combined intervention - the evidence for known mechanisms and efficaciousness?
Reviewer 3 Report
Comments and Suggestions for Authors
The authors provide an overview of non-invasive brain stimulation (NIBS), and discuss how it can be used to manage chronic musculoskeletal pain (CMP). The authors conclude that while the neural mechanisms behind pain reduction are plausible, many of the studies in this field are poorly designed, which means that it is difficult to evaluate the effectiveness of the treatment. There is currently a lack of high-quality evidence to show that NIBS is good treatment for CMP.
Feedback:
1) The results could be presented better. There is only 1 figure in the paper, and while it's fine, it isn't really necessary to demonstrate the conclusions presented. I think more figures are needed, which are able to tell the reader at a glance what the authors are trying to say in that section.
E.g. "There is a pressing need for rigorously designed, adequately powered, and properly blinded RCTs". Perhaps there could be a figure, showing how many studies on NIBS for CMP have been done in the last 10 years, and out of those what % were rigorously designed, what % were adequately powered, and what % were blinded?
This should also be done for sections 4.1/4.2/4.3/4.4. Either a figure or a table to summarise the findings. It's very hard to keep track of multiple numbers including mean results, standardised mean differences, confidence intervals, and p-values buried in text.
2) As this paper is in itself a kind of meta-analysis, it would be good to implement standardised reporting guidelines such as PRISMA.
Round 2
Reviewer 3 Report
Comments and Suggestions for Authors
Thank you.